# Genotypic Variation in Nickel Accumulation and Translocation and Its Relationships with Silicon, Phosphorus, Iron, and Manganese among 72 Major Rice Cultivars from Jiangsu Province, China

**DOI:** 10.3390/ijerph16183281

**Published:** 2019-09-06

**Authors:** Ya Wang, Chengqiao Shi, Kang Lv, Youqing Li, Jinjin Cheng, Xiaolong Chen, Xianwen Fang, Xiangyang Yu

**Affiliations:** 1Jiangsu Key Laboratory for Food Quality and Safety-State Key Laboratory Cultivation Base, Ministry of Science and Technology, Nanjing 210014, China (Y.W.) (C.S.) (K.L.) (Y.L.) (J.C.) (X.C.); 2Institute of Food Safety and Nutrition, Jiangsu Academy of Agricultural Sciences, Nanjing 210014, China; 3National Crop Germplasm Resources Infrastructure (Jiangsu), Ministry of Science and Technology, Nanjing 210014, China

**Keywords:** *Oryza sativa* L., toxic element, redundancy analysis, cultivar variation

## Abstract

Nickel (Ni) is a ubiquitous environmental toxicant and carcinogen, and rice is a major dietary source of Ni for the Chinese population. Recently, strategies to decrease Ni accumulation in rice have received considerable attention. This study investigated the variation in Ni accumulation and translocation, and also multi-element (silicon (Si), phosphorus (P), iron (Fe), and manganese (Mn)) uptake and transport among 72 rice cultivars from Jiangsu Province, China, that were grown under hydroponic conditions. Our results showed a 2.2-, 4.2-, and 5.3-fold variation in shoot Ni concentrations, root Ni concentrations, and translocation factors (TFs) among cultivars, respectively. This suggests that Ni accumulation and translocation are significantly influenced by the genotypes of the different rice cultivars. Redundancy analysis of the 72 cultivars revealed that the uptake and transport of Ni were more similar to those of Si and Fe than to those of P and Mn. The Ni TFs of high-Ni cultivars were significantly greater than those of low-Ni cultivars (*p* < 0.001). However, there were no significant differences in root Ni concentrations of low-Ni and high-Ni cultivars, suggesting that high-Ni cultivars could translocate Ni to shoots more effectively than low-Ni cultivars. In addition, the cultivars HD8 and YD8 exhibited significantly lower levels of Ni accumulation than their parents (*p* < 0.05). Our results suggest that breeding can be an effective strategy for mitigating excessive Ni accumulation in rice grown in Ni-contaminated environments.

## 1. Introduction

Nickel (Ni) is a ubiquitous trace metal that has both natural and anthropogenic sources (e.g., vehicle emissions, as well as the Ni mining, smelting, cement manufacture, metallurgical, and electroplating industries), and its environmental accumulation has become a concern worldwide [1,2]. In recent years, fertilizer and organic manure use have increased Ni concentrations in cropland soils, exacerbating the problem of Ni pollution [2,3]. A recent nationwide Chinese soil survey revealed that 19.4% of the cropland soil samples were polluted. Ni was the second most abundant (4.8%) potentially toxic element (PTE) in soil and a key pollutant of Chinese farmlands [4]. In Jiangsu Province, located in the eastern coastal region of China, the surface soil (0–20 cm) background concentrations of Ni varied from 1.6 to 238 mg·kg^−1^, with a geometric mean of 32.9 mg·kg^−1^ [5]. Although the average soil Ni concentration in Jiangsu is below the 40 mg·kg^−1^ class II standard described by the environmental quality standard for soils of China (GB 15618–1995), the level of Ni pollution cannot be ignored, especially in southern Jiangsu where the level of anthropogenic contamination has increased [5,6]. In fact, a geological survey found that among all soils polluted with PTEs in Jiangsu Province, Ni-polluted soils were the most common (2.91% of all agricultural soils tested) [5]. The recent increase in the levels of Ni soil pollution and its implications for adverse effects on human health have focused attention on the processes of Ni accumulation and translocation in cereal crops [3,7,8,9].

Rice (*Oryza sativa* L.) is a staple food for approximately half of the global population [10,11]. However, rice grains also accumulate PTEs (e.g., cadmium (Cd), arsenic (As), and Ni) efficiently [8,12,13,14]. In contrast to PTEs such as Cd and As, Ni is an essential micronutrient that is required for plant growth because it is a key component of the enzyme urease [3,7,15]. However, at high levels, Ni is toxic to rice and can inhibit seed germination [16], suppress growth and reduce biomass [7], decrease the quantity of photosynthetic pigments [7], stimulate lipid peroxidation [15], and disrupt carbohydrate metabolism [17]. Although Ni is essential for plant growth, it has no proven biochemical function in humans and may be unnecessary [8,9]. Furthermore, the presence of Ni may increase the likelihood of cancer, heart attacks, skin problems, vomiting, and respiratory illnesses [8,9,18]. Consequently, the European Union has implemented a maximum permitted level of 75 μg Ni day^−1^ in the diet [19]. However, Ni is one of the most abundant PTEs found in rice [9,13], and its geometric mean varies from 0.46 to 0.54 mg·kg^−1^ in the three main rice-producing regions of China [20]. This is a much higher Ni concentration than is typical of rice from other countries [13,21]. Consequently, rice is a major dietary source of Ni for the Chinese population, especially individuals who consume a lot of rice.

Food safety strategies to reduce Ni assimilation into rice are considered important but have received less attention than those focusing on other PTEs such as lead (Pb), mercury (Hg), As, Cd, and chromium (Cr) [9,10,11,12,14]. Phytoremediation and soil flushing techniques are promising methods for the remediation of PTE-contaminated soils [22]. However, serious limitations such as low biomass, propagation difficulties, and the time and expense involved have prevented these methods from being used on a wider scale [2,22]. Recent research has demonstrated that inter-cultivar variation plays a significant role in determining PTE concentrations in rice grains [23,24]. Therefore, selecting cultivars that restrict Ni translocation and accumulation in rice grains may be the simplest cost-effective approach to preventing excessive Ni accumulation. Interestingly, nutrient elements such as silicon (Si) and phosphorus (P) can affect the growth and yield of rice [25,26]. Furthermore, they can influence the uptake of PTEs, such as Cd and As [27,28,29,30,31,32,33]. However, it is unclear whether Ni translocation and accumulation can be influenced by Si and P in different rice subgroups. In addition, many studies have demonstrated that iron (Fe) and manganese (Mn) play important roles in mediating the accumulation of PTEs (e.g., As, Cd, Pb, and Ni) in rice through the formation of plaque deposits on the roots [33,34,35,36,37,38,39]. Nevertheless, the relationships between the uptake and translocation of Ni and other elements may differ among rice genotypes, and further studies are needed to understand these relationships.

In this study, we performed short-term Ni uptake experiments with 72 major rice cultivars that were grown under hydroponic conditions. The objectives of the study were to (1) evaluate the variation in Ni accumulation and translocation among 72 rice cultivars, (2) analyze the relationships among Ni and multi-element (Si, P, Fe, and Mn) uptake and translocation among the rice cultivars, and (3) identify the major factors that affect the accumulation and translocation of Ni in different rice genotypes.

## 2. Materials and Methods

### 2.1. Rice Cultivars and Growth Conditions

Table 1 lists the 72 major rice (*O*. *sativa* L.) cultivars used in this study, including 64 *japonica* cultivars and 8 *indica* cultivars (YD1–8). These cultivars were obtained from the national crop germplasm resources infrastructure (Jiangsu, China), the Ministry of science and technology (Beijing, China), and the Jiangsu academy of agricultural sciences (Nanjing, China). Rice seeds were surface-sterilized in 30% (v/v) hydrogen peroxide for 15 min, washed three times in sterile deionized water [40], and then transferred to an incubator for germination in the dark at 32 °C for 2 days. Pre-germinated rice seedlings were grown hydroponically in a nutrient solution, in accordance with the method described by the international rice research institute under the following conditions: 12 h light/12 h dark cycle (light fluence rate, 360 μmol m^−2^·s^−1^) at 30 °C/25 °C in an environmental chamber with a relative humidity of 60–70% [41]. The seedlings were cultured in sterile deionized water before the one-leaf stage, and in 0.25- and 0.5-strength nutrient solutions at the two- and three-leaf stages, respectively. The nutrient solutions were renewed every 3 days, and the pH of the solutions was maintained at 5.6 with 5 mmol·L^−1^ 2-(N-morpholino) ethanesulfonic acid [41,42].

### 2.2. Experimental Design

Uniform 20-day-old rice seedlings at the three-leaf stage were selected and cultured in 400-mL plastic cups. Each cup contained three seedlings and 350 mL of 0.5-strength nutrient solution. The seedlings were treated with 0 and 10 μmol·L^−1^ Ni [7], and each treatment was replicated three times. A Ni stock solution was prepared from nickel sulfate hexahydrate (NiSO_4_·6H_2_O). After exposure to 10 μmol·L^−1^ Ni for 3 days (short-term experiment), the rice samples were collected, rinsed three times with deionized water, and then separated into shoots and roots. In addition, 2 mL aliquots of the nutrient solution from each replicate were filtered using a 0.45-μm syringe filter and then stored at 4 °C to determine the total Ni concentrations. After oven-drying the sample tissues for 2 days at 60 °C, the dry weights of the shoots and roots were recorded. Then, the tissues were ground using zirconia beads in a high-throughput sample grinder (CK-2000; Thmorgan, Beijing, China). The translocation factor (TF) was calculated as the shoot Ni concentration/root Ni concentration. The bioconcentration factor (BCF) of Ni from the culture medium to shoots or roots was calculated as follows: BCF = shoot or root Ni concentration/Ni concentration in the medium.

### 2.3. Sample Analyses

To analyze the total Si, Ni, P, Fe, and Mn concentrations in the rice tissues, approximately 0.05 g of shoot or 0.01 g of root tissue was weighed into 50 mL centrifuge tubes (Crystalgen, Inc., Commack, NY, USA). A total of 5 mL of concentrated HNO_3_ was added to each tube and incubated overnight at 20 °C. The samples were digested using a digital block digestion system (ED54 DigiBlock; LabTech, Beijing, China) at 120 °C for 2 h [40]. After cooling, the samples were diluted to 50 mL using Milli-Q water (18.2 MΩ·cm^−1^; Merck Millipore, Burlington, MA, USA). The elements described above were identified using inductively coupled plasma mass spectrometry (ICP-MS; Nexion 350D; Perkin Elmer, Waltham, MA, USA). A total of 20 μg·L^−1^ of scandium and germanium were used as internal standards. Rice flour (NIST-SRM 1568b) and GBW10010 (GSB-1) were also digested and analyzed as reference materials to ensure the results were accurate [40]. The measured values for total P (1551 ± 17 mg·kg^−1^), Fe (7.40 ± 0.87 mg·kg^−1^), and Mn (19.5 ± 0.153 mg·kg^−1^) in SRM 1568b were similar to the certified values (1530 ± 40 mg·kg^−1^, 7.42 ± 0.44 mg·kg^−1^, and 19.2 ± 1.8 mg·kg^−1^, respectively). The measured values for total Ni (0.245 ± 0.012 mg·kg^−1^) and Si (231 ± 22 mg·kg^−1^) in GSB-1 were lower than the certified values but within acceptable limits (0.27 ± 0.02 mg·kg^−1^ and 250 ± 30 mg·kg^−1^, respectively). The Ni concentrations of the nutrient solutions were also determined using ICP-MS, with scandium as an internal standard.

### 2.4. Statistical Analyses

SigmaPlot software (ver. 12.5; Systat, San Jose, CA, USA) was used to create the figures. Independent samples *t*-test, one-way analysis of variance, and post-hoc multiple comparisons (Tukey’s test) were used to determine the significance (*p* < 0.05 or 0.01) of the results using SPSS software (ver. 20.0; IBM Corp., Armonk, NY, USA). Pearson’s correlation analysis was also performed using SPSS (ver. 20.0). Redundancy analysis (RDA) was performed to analyze the relationships among the accumulation and translocation of Ni and other elements in the rice cultivars using Canoco software (ver. 4.5; Microcomputer Power, Ithaca, NY, USA) [43].

## 3. Results

### 3.1. Accumulation and Translocation of Ni in 72 Rice Cultivars

As shown in Figure 1, we investigated the genotypic variation among 72 rice cultivars in shoot and root Ni concentrations and root-to-shoot Ni translocation. After exposure to 10 μmol·L^−1^ Ni for 72 h, significant differences were observed among the rice cultivars.

Shoot Ni concentrations varied from 13.3 mg·kg^−1^ in HD5 to 29.1 mg·kg^−1^ in the early rice LJ6 (geometric mean: 22.1 mg·kg^−1^, median: 22.1 mg·kg^−1^; Figure 1A). Root Ni concentrations varied from 385 mg·kg^−1^ in YD8 to 1602 mg·kg^−1^ in NJ34 (geometric mean: 715 mg·kg^−1^, median: 703 mg·kg^−1^; Figure 1B). The TFs of Ni from rice roots to shoots ranged from 0.011 (SJ4) to 0.058 (YD8). The translocation of Ni in rice cultivar YD8 was 5.3-fold greater than in SJ4 (Figure 1C).

The BCFs were calculated to investigate the Ni accumulation capacities of shoots and roots among the rice cultivars (Appendix A). In general, the BCFs of shoots and roots differed significantly among the cultivars. The shoot Ni BCFs varied from 17.8 to 38.8, whereas the root Ni BCFs varied from 513 to 2136. HD5 had the lowest and LJ6 (early rice) had the highest shoot Ni BCFs (Appendix A). YD8 had the lowest and NJ34 had the highest root Ni BCFs (Appendix A).

### 3.2. Variation in Shoot Ni Concentrations among Different Rice Subgroups

Table 2 shows the 72 rice cultivars separated into 11 subgroups based on their locations within Jiangsu Province. To characterize the variation in Ni accumulation among these subgroups, we defined the 20 lowest and 20 highest shoot Ni-accumulating genotypes among the 72 cultivars as low-Ni and high-Ni cultivars, respectively. Most of the low-Ni cultivars were found in four subgroups (i.e., HD, LJ, W–J, and YJ; Table 2). In addition, the proportions of high-Ni cultivars in these four subgroups were lower compared to the proportions of low-Ni and mid-Ni cultivars (22.2%, 22.2%, 25%, and 25% of the total cultivars, respectively). Furthermore, most of the cultivars in the TJ subgroup were low-Ni cultivars, and this subgroup also had the lowest geometric mean (18.9 mg·kg^−1^). In contrast, there were no low-Ni cultivars in the XD subgroup, which had the highest geometric mean (24.5 mg·kg^−1^).

We also compared the shoot Ni concentrations of the rice cultivars with those of their parents (Table 3). The shoot Ni concentrations of HD11 and ZD16 were similar to those of their parents. In contrast, the shoot Ni concentrations of HD8 and YD8 were significantly lower than those of their parents (*p* < 0.05), whereas the shoot Ni concentration of YJ2 was significantly higher than that of its parent (*p* < 0.05).

### 3.3. Relationships between Accumulation and Translocation of Si, P, Fe, Mn, and Ni in the Rice Cultivars

Shoot Ni concentrations were positively correlated with Ni, P, and Fe TFs (*p* < 0.01), but negatively correlated with root Fe concentrations (*p* < 0.01) (Table 4). Root Ni concentrations were positively correlated with Mn TFs (*p* < 0.05), but negatively correlated with Ni TFs (*p* < 0.01), shoot Fe (*p* < 0.01), and Si concentrations in rice shoots and roots (*p* < 0.05). The correlation analysis also found a positive relationship between Ni TFs and P TFs, and among the concentrations of Si, Ni, and Fe in shoots (*p* < 0.01), but a negative relationship between Ni TFs and root Ni concentrations (*p* < 0.01).

Among the 20 low-Ni cultivars, we observed a positive relationship between shoot Ni concentrations and Ni TFs (*p* < 0.05; Appendix A). However, there was no such relationship among the 20 high-Ni cultivars (Appendix A). In these high-Ni cultivars, shoot Ni concentrations were positively correlated with Mn TFs (*p* < 0.05). Root Ni concentrations were negatively correlated with Ni TFs in both low-Ni and high-Ni cultivars (*p* < 0.01). In addition, root Ni concentrations were positively correlated with Fe TFs and Mn TFs in low-Ni and high-Ni cultivars, respectively (both, *p* < 0.05; Appendix A). The Ni TFs were positively correlated with shoot Si concentrations in both low-Ni and high-Ni cultivars (*p* < 0.05). Furthermore, the Ni TFs were positively correlated with shoot Fe concentrations in high-Ni cultivars (*p* < 0.05), but the correlation coefficients were not significant for the low-Ni cultivars.

### 3.4. Relationships among the Accumulation and Translocation of Ni and Other Elements in the Rice Subgroups

Both the first canonical axis and all canonical axes explained a significant amount of the variation (*p* = 0.004 and *p* = 0.002, respectively) based on the Monte Carlo permutation test (number of permutations = 499; Figure 2). The first and second axes contributed 29.1% and 9.7% of the total variation, respectively. For Ni accumulation (displayed as shoot Ni and root Ni concentrations) and translocation (displayed as Ni TFs), shoot Si (*p* = 0.002; F = 14.39) and root Fe (*p* = 0.008; F = 6.00) concentrations accounted for the variation among the 72 rice cultivars. Shoot Si concentrations explained the greatest proportion of this variation (17%), followed by root Fe concentrations (7%).

An RDA of the relationships among accumulation and translocation of Ni and multi-element concentrations in the 20 lowest and 20 highest Ni-accumulating rice cultivars is shown in Figure 3. The high-Ni rice cultivars mainly clustered in the left upper and lower quadrants, whereas the low-Ni rice cultivars clustered in the lower right and left quadrants. The first and second RDA axes accounted for 21.9% and 17.0% of the total variation, respectively (both, *p* = 0.002). The shoot Si and P TFs accounted for 12% and 15% of the Ni accumulation and translocation variation, respectively (both, *p* < 0.01).

The first and second axes accounted for 75.7% and 4.5% of the total variation among the 20 lowest Ni-accumulating rice cultivars, respectively (*p* = 0.002; Appendix A). The Si TFs, root Fe and shoot Fe concentrations, and P TFs accounted for 40%, 11%, 9%, and 6% of the variation in Ni accumulation and translocation, respectively (all, *p* < 0.05). In contrast, an RDA analysis of the relationships among Ni accumulation, translocation and multi-element concentrations in the 20 high-Ni rice cultivars is not shown because the Monte Carlo tests of the first canonical axis and all canonical axes were not significant (*p* = 0.218 and *p* = 0.23, respectively).

### 3.5. Effects of Ni Exposure on Rice Seedling Growth

We also investigated the effects of Ni exposure on rice growth, under hydroponic conditions. The five lowest and highest shoot Ni accumulating genotypes were selected from among the 72 cultivars, and the effect on biomass (dry weight) of providing 10 μmol·L^−1^ Ni was analyzed (Appendix A). No significant differences in shoot (Appendix A) or root (Appendix A) biomass were observed in any of the ten cultivars. Similar results were observed for the remaining 62 cultivars (data not shown). These results suggest that exposure to Ni for 3 days did not significantly affect rice seedling growth.

### 3.6. Effects of Ni Exposure on Multi-Element Uptake

To further investigate the effects of Ni exposure on the uptake of nutrients, we analyzed the presences of multiple elements in rice shoots and roots after exposure to 10 μmol·L^−1^ Ni for 3 days. In general, the uptake of Si, P, Fe, and Mn in rice shoots and roots decreased in response to Ni exposure. In addition, Ni exposure had a much greater effect on multi-element uptake in the shoots of the five highest Ni-accumulating cultivars than in those of the five lowest Ni-accumulating cultivars (Appendix A). Among the low-Ni cultivars, the uptakes of Si and Fe were considerably more affected by Ni exposure than those of P and Mn (Appendix A). In addition, the Mn uptake in shoots and roots of the five high-Ni cultivars decreased significantly (Appendix A). Furthermore, in roots exposed to Ni, Fe concentrations were much higher than Mn concentrations in both low-Ni and high-Ni cultivars. Conversely, in rice shoots, Mn concentrations were much higher than Fe concentrations (Appendix A).

## 4. Discussion

Ni is the major PTE pollutant present in rice grains [9,20], and rice is a major dietary source of Ni for the Chinese population, particularly children aged 2–11 years and Ni-sensitive individuals [20]. However, strategies to decrease Ni accumulation in rice have received little attention until now. Many studies have shown that the concentrations of PTEs (e.g., Cd, As, Pb, and Ni) in grains from various rice subpopulations differ significantly [12,20,44,45]. Therefore, it may be possible to identify cultivars that accumulate significantly lower levels of Ni. In China, approximately 7.4% of the total rice-planting area is located in Jiangsu Province and 90% of the rice grown is of the *japonica* variety [46]. In this study, we investigated Ni accumulation and translocation in 72 major rice cultivars, as well as multi-element uptake and translocation in different rice tissues. Our results may be used to prevent excessive Ni accumulation in rice grown in Ni-contaminated soil and to improve food safety.

There were significant differences in Ni accumulation and translocation across the 72 major rice cultivars after 3 days of Ni exposure (Figure 1). Shoot Ni concentrations were positively correlated with Ni TFs (*p* < 0.01), but not with root Ni in all rice cultivars (Table 4). Therefore, differences in shoot Ni concentrations were explained by the different Ni-transport capacities of the rice genotypes rather than the immobilization of Ni in roots in response to Ni exposure. Previous studies have shown that phytochelatins (PCs) are important for PTE detoxification in plants [47,48]. However, in contrast to responses induced by PTEs such as Cd and As (including arsenite and trivalent methylarsonous acid) [48], PC synthesis was not strongly induced by Ni [47,49], suggesting that PC synthesis and subsequent Ni sequestration in roots are less important for Ni detoxification. Similar results were also observed in low-Ni (20 lowest shoot Ni BCFs; Appendix A) and high-Ni (20 highest shoot Ni BCFs; Appendix A) cultivars. The geometric means of shoot Ni concentrations (26.5 ± 1.33 mg·kg^−1^) and Ni TFs (0.037 ± 0.006) in high-Ni cultivars were significantly (*p* < 0.001) greater than those of low-Ni cultivars (18.0 ± 1.99 mg·kg^−1^ and 0.025 ± 0.007, respectively; Figure 1A,C). However, the root Ni concentrations of low-Ni (792 ± 278 mg·kg^−1^) and high-Ni cultivars (738 ± 142 mg·kg^−1^) were not significantly different (Figure 1B). These results suggest that the high-Ni cultivars translocate Ni to shoots more effectively than low-Ni cultivars. In addition, although root Ni sequestration is not the most important factor affecting shoot Ni concentrations, root Ni concentrations were negatively correlated with Ni TFs in both low-Ni and high-Ni cultivars (*p* < 0.01; Appendix A). These observations suggest that cultivars that sequester greater concentrations of Ni in their roots can decrease Ni translocation to their shoots.

To further identify the major factors that affect the uptake and translocation of Ni in different rice genotypes, RDA was used to analyze the relationships among the accumulation and translocation of Ni and other elements in rice tissues (Figure 2, Figure 3 and Appendix A). The results showed that shoot Si and root Fe concentrations significantly affected Ni accumulation and translocation in the 72 cultivars studied (*p* < 0.01; Figure 2). Therefore, the uptake and transport of Ni is closely associated with Si and Fe concentrations. Si is an important nutrient for rice growth [25,50] and also protects plants from toxic metals such as Cd, As, Ni, and Zn by enhancing growth and photosynthetic carbon fixation, and suppressing the uptake of toxic metals [31,51,52,53]. Therefore, rice cultivars with a greater capacity for Si assimilation may also accumulate less Ni in their shoots and roots. In addition, the geometric mean of root Si concentrations was significantly greater in the five low-Ni cultivars than in the five high-Ni cultivars, regardless of whether the plants were exposed to Ni (*p* < 0.05; Appendix A). Furthermore, root Fe concentrations were correlated with Ni uptake and transport in rice (*p* < 0.01; Figure 2 and Appendix A). Interestingly, Fe plaque deposits, which are visible as a reddish-brown coating on the surface of roots, can sequestrate PTEs such as Cd, As, and Ni and reduce their toxicity [34,38,54]. However, we found no evidence of Ni and Fe co-precipitation on the surface of roots, and no significant correlations between root Fe and root Ni concentrations were observed (Table 4, Appendix A). The excess P in the culture medium (i.e., 5 mg L^−1^ P for 3 days) might be responsible for the results because previous studies have demonstrated that iron plaque is induced by P starvation [33]. Nonetheless, Fe accumulation in rice roots was correlated with the uptake and transport of Si in the 72 cultivars (*p* < 0.01; Table 4), and this may have an indirect effect on the uptake and transport of Ni in rice. Similarly, indirect effects of P TFs on the uptake and transport of Ni among 20 low-Ni and high-Ni cultivars were also observed (Figure 3), because P TFs were correlated with the uptake and transport of Si and Fe but not Ni (Appendix A).

Our results indicated that the uptake of nutrient elements was inhibited in rice under Ni stress. Appendix A show that the concentrations of most nutrient elements in rice shoots and roots decrease after exposure to Ni for 3 days in both low-Ni and high-Ni cultivars. Differences in shoot or root element concentrations in response to Ni exposure were probably due to Ni rather than a difference in rice growth, because no differences were observed in shoot (Appendix A) or root (Appendix A) biomass after 72 h of Ni exposure. Similar results were obtained from other rice studies, which showed decreased K, Ca, Mg, Fe, and Mn uptake and distribution [49,55,56]. This may be due to alterations in root membrane permeability in response to excessive exposure to Ni [55]. Furthermore, our data suggest that the high-Ni rice cultivars were much more sensitive to Ni than the low-Ni cultivars, especially with respect to accumulation of Si and Mn in rice shoots (Appendix A). This may be crucial for the enhanced shoot Ni accumulation in high-Ni cultivars because these nutrients are essential for rice growth and are also important for alleviating PTEs toxicity [31,57].

We showed that genotype has a significant effect on shoot Ni concentrations among the different rice subgroups (Table 2). In addition, significant differences among cultivars were found within the same subgroup. For example, among the 72 cultivars, those with the lowest (HD5 and LJ6) and highest (HD10 and LJ6: early rice) concentrations of Ni in their shoot were found in the same subgroups (HD and LJ). Genetic differences in the parent plants may be responsible for differences in the PTE accumulation capacities of different rice cultivars [20,58]. However, it is unclear how genotype affects Ni accumulation. Some studies have suggested that progeny plants inherit genes that decrease the accumulation of PTEs from parent plants that also show low levels of PTE accumulation [59,60]. Although cultivar YJ2 exhibited a greater capacity for Ni accumulation than its parent, other cultivars (e.g., HD8 and YD8) exhibited a similar capacity for decreased Ni accumulation to their parents (Table 3), indicating that it should be possible to breed rice that accumulates lower levels of Ni. The four subgroups with the greatest number of low-Ni cultivars (i.e., HD, LJ, W–J, and YJ) are shown in Table 2 and these may be used to further investigate the genetics underpinning low-Ni accumulation.

## 5. Conclusions

This study demonstrates that genotype had a significant effect on Ni accumulation and translocation in a population of 72 rice cultivars. The variation in the shoot Ni concentrations was explained by different capacities for Ni transport in different rice genotypes rather than by the immobilization of Ni in roots exposed to high Ni concentrations. In general, the Ni TFs of high-Ni cultivars were significantly greater than those of low-Ni cultivars. The RDA of the 72 rice genotypes suggested that the uptake of Si and Fe was the major factor affecting the accumulation and translocation of Ni. However, significant differences were also observed between 20 low-Ni and 20 high-Ni rice cultivars. Among the 20 low-Ni cultivars, Si TFs accounted for most of the variation in Ni accumulation and translocation. However, P TFs accounted for most of the variation in Ni uptake and translocation observed between the 20 low-Ni and 20 high-Ni cultivar groups. Mn was less important than Si, Fe, and P in influencing Ni accumulation and translocation in these cultivars. The results of this study may be used to prevent excessive Ni accumulation in rice grown in Ni-contaminated environments. Further studies are needed to investigate how genotypes influence Ni uptake and transport in rice.

## Figures and Tables

**Figure 1 ijerph-16-03281-f001:**
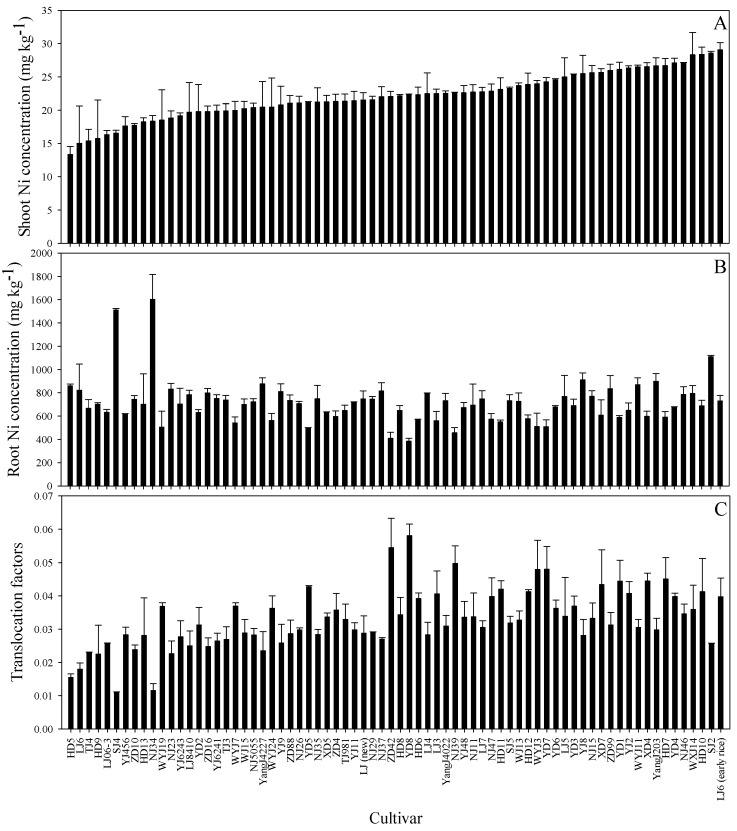
Nickel (Ni) concentrations in shoots (**A**), roots (**B**), and translocation factors (**C**) of 72 different rice seedlings. Data are means ± standard deviation (n = 3).

**Figure 2 ijerph-16-03281-f002:**
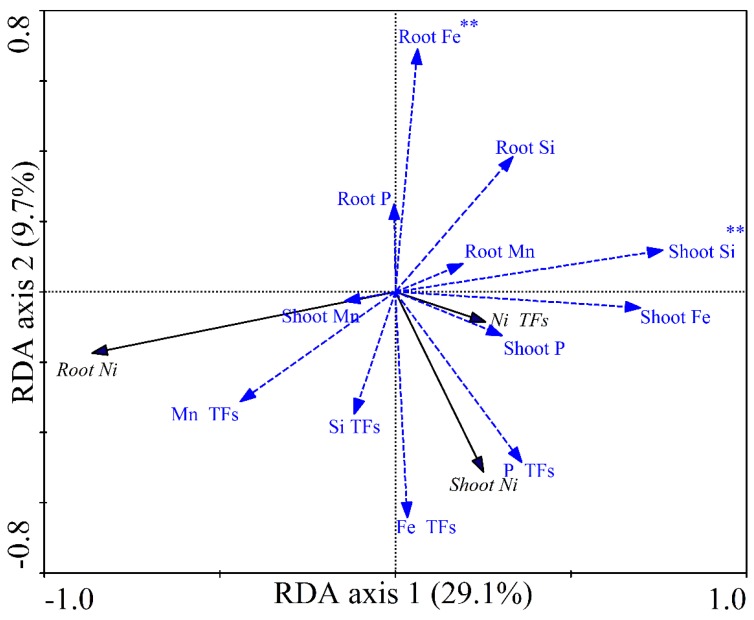
RDA (Redundancy analysis) ordination diagrams of the relationships between accumulation and translocation of Ni and multi-element concentrations in 72 rice cultivars. Shoot Ni, root Ni, and Ni TFs (translocation factors) are represented by black lines with arrows. Shoot Si, shoot P, shoot Fe, shoot Mn, root Si, root P, root Fe, root Mn, Si TFs, P TFs, Fe TFs, and Mn TFs are represented by blue lines with arrows. ** (*p* < 0.01) represent significant factors influencing Ni accumulation and translocation based on Monte Carlo analysis (the number of permutations = 499).

**Figure 3 ijerph-16-03281-f003:**
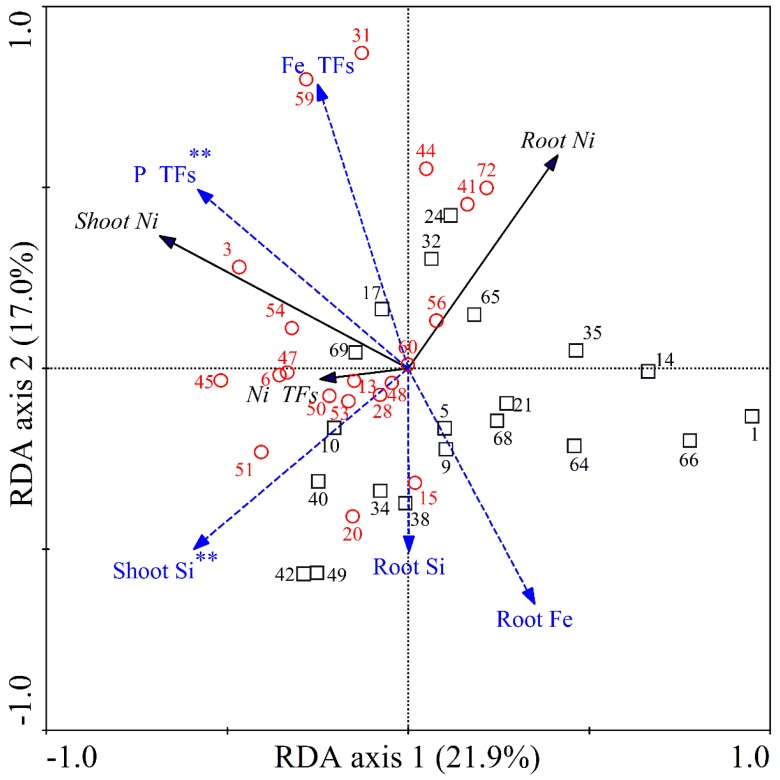
RDA ordination diagrams of the relationships among Ni and multi-element uptake and translocation in 40 rice cultivars. Shoot Ni, root Ni, and Ni TFs are displayed as black lines with arrows. Shoot Si, root Si, root Fe, P TFs, and Fe TFs are represented by blue lines with arrows. The 20 lowest Ni-accumulating rice cultivars are represented by squares “□”. The 20 highest Ni-accumulating rice cultivars are represented by circles “○”. The numbers around the squares or circles represent 40 associated rice cultivars listed in Table 1. ** (*p* < 0.01) represent significant factors influencing Ni accumulation and translocation based on Monte Carlo analysis (the number of permutations = 499).

**Table 1 ijerph-16-03281-t001:** Rice cultivars from Jiangsu Province.

Number	Name	Abbreviation	Number	Name	Abbreviation
1	Huaidao 5	HD5	37	Wujing 13	WJ13
2	Huaidao 6	HD6	38	Wujing 15	WJ15
3	Huaidao 7	HD7	39	Wuyujing 3	WYJ3
4	Huaidao 8	HD8	40	Wuyunjing 7	WYJ7
5	Huaidao 9	HD9	41	Wuyunjing 11	WYJ11
6	Huaidao 10	HD10	42	Wuyunjing 19	WYJ19
7	Huaidao 11	HD11	43	Wuyunjing 24	WYJ24
8	Huaidao 12	HD12	44	Wuxiangjing 14	WXJ14
9	Huaidao 13	HD13	45	Xudao 4	XD4
10	Lianjing 06-3	LJ06-3	46	Xudao 5	XD5
11	Lianjing 3	LJ3	47	Xudao 7	XD7
12	Lianjing 4	LJ4	48	Yangdao 1	YD1
13	Lianjing 5	LJ5	49	Yangdao 2	YD2
14	Lianjing 6	LJ6	50	Yangdao 3	YD3
15	Lianjing 6 (early rice)	LJ6 (early rice)	51	Yangdao 4	YD4
16	Lianjing 7	LJ7	52	Yangdao 5	YD5
17	Lianjing 8410	LJ8410	53	Yangdao 6	YD6
18	Lianjing (new)	LJ (new)	54	Yangdao 7	YD7
19	Nanjing 11	NJ11	55	Yangdao 8	YD8
20	Nanjing 15	NJ15	56	Yangjing 203	YangJ203
21	Nanjing 23	NJ23	57	Yangjing 4022	YangJ4022
22	Nanjing 26	NJ26	58	Yangjing 4227	YangJ4227
23	Nanjing 29	NJ29	59	Yanjing 2	YJ2
24	Nanjing 34	NJ34	60	Yanjing 8	YJ8
25	Nanjing 35	NJ35	61	Yanjing 9	YJ9
26	Nanjing 37	NJ37	62	Yanjing 11	YJ11
27	Nanjing 39	NJ39	63	Yanjing 48	YJ48
28	Nanjing 46	NJ46	64	Yanjing 456	YJ456
29	Nanjing 47	NJ47	65	Yanjing 6241	YJ6241
30	Nanjing 5055	NJ5055	66	Yanjing 6243	YJ6243
31	Sujing 2	SJ2	67	Zhendao 4	ZD4
32	Sujing 4	SJ4	68	Zhendao 10	ZD10
33	Sujing 5	SJ5	69	Zhendao 16	ZD16
34	Tongjing 3	TJ3	70	Zhendao 42	ZD42
35	Tongjing 4	TJ4	71	Zhendao 88	ZD88
36	Tongjing 981	TJ981	72	Zhendao 99	ZD99

**Table 2 ijerph-16-03281-t002:** Ni concentrations in shoots of rice subgroups from Jiangsu Province.

Cultivars	Number of Cultivars	Minimum (mg·kg^−1^)	Maximum (mg·kg^−1^)	Mean (mg·kg^−1^)	Number of Cultivars and Proportion (%)		
					Low-Ni	Mid-Ni	High-Ni
HD	9	13.3 ± 1.20	28.4 ± 1.28	21.5	3 (33.3)	4 (44.4)	2 (22.2)
LJ	9	15.0 ± 5.58	29.1 ± 1.17	21.6	3 (33.3)	4 (44.4)	2 (22.2)
NJ	12	18.3 ± 0.864	27.1 ± 0.045	22.0	2 (16.7)	8 (66.7)	2 (16.7)
SJ	3	16.6 ± 0.425	28.6 ± 0.289	22.8	1 (33.3)	1 (33.3)	1 (33.3)
TJ	3	15.4 ± 1.71	21.4 ± 1.08	18.9	2 (66.7)	1 (33.3)	–
W–J	8	18.5 ± 4.55	28.3 ± 3.37	22.7	3 (37.5)	3 (37.5)	2 (25)
XD	3	21.3 ± 0.958	26.5 ± 0.580	24.5	–	1 (33.3)	2 (66.7)
YD	8	19.8 ± 4.08	27.1 ± 0.687	23.8	1 (12.5)	5 (62.5)	2 (25)
YangJ	3	20.4 ± 3.84	26.7 ± 1.19	23.2	–	2 (66.7)	1 (33.3)
YJ	8	17.6 ± 1.39	26.4 ± 0.338	21.7	3 (37.5)	3 (37.5)	2 (25)
ZD	6	17.7 ± 0.267	26.0 ± 0.905	21.3	2 (33.3)	3 (50)	1 (16.7)

Note: The low Ni-accumulating (low-Ni) and high Ni-accumulating (high-Ni) cultivars are defined as the 20 lowest and 20 highest shoot Ni-accumulating genotypes among the 72 cultivars, respectively. The other 32 cultivars are defined as middle Ni-accumulating (mid-Ni) cultivars. The subgroup W–J comprised two WJ, five WYJ, and one WXJ cultivars. Data are means ± standard deviation (n = 3).

**Table 3 ijerph-16-03281-t003:** The Ni concentration in shoots of different rice cultivars and their parents.

Number	Cultivars Abbreviation	Shoot Ni Concentration (mg·kg^−1^)
1	Cultivar	HD8	22.2 ± 0.190 b
Maternal line ( 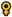 )	WYJ3	24.0 ± 0.487 a
Paternal line ( 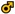 )	WYJ3	24.0 ± 0.487 a
2	Cultivar	HD11	23.1 ± 1.736 a
Maternal line ( 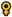 )	HD9	15.7 ± 5.78 a
Paternal line ( 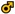 )	HD9	15.7 ± 5.78 a
3	Cultivar	YJ2	26.4 ± 0.338 a
Maternal line ( 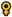 )	NJ11	22.7 ± 1.10 b
Paternal line ( 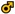 )	NJ11	22.7 ± 1.10 b
4	Cultivar	ZD16	19.8 ± 0.861 a
Maternal line ( 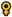 )	ZD88	21.0 ± 1.15 a
Paternal line ( 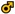 )	WJ15	20.2 ± 1.04 a
5	Cultivar	ZD99	26.0 ± 0.905 a
Maternal line ( 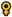 )	ZD88	21.0 ± 1.15 b
Paternal line ( 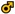 )	WYJ3	24.0 ± 0.487 a
6	Cultivar	YD8	22.3 ± 0.128 b
Maternal line ( 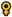 )	YD6	24.6 ± 0.15 a
Paternal line ( 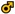 )	YD6	24.6 ± 0.15 a

Note: Different lowercase letters denote significantly different at *p* < 0.05 between cultivar and its parents according to a Tukey’s multiple comparison test. Data are means ± standard deviation (n = 3).

**Table 4 ijerph-16-03281-t004:** Pearson correlation coefficients between accumulation and translocation of multi-element in 72 rice cultivars.

		Shoot Concentrations					Root Concentrations					TFs				
		Si	Ni	P	Fe	Mn	Si	Ni	P	Fe	Mn	Si	Ni	P	Fe	Mn
Shoot concentrations	Si	1	0.108	0.408 **	0.687 **	0.070	0.565 **	−0.427 **	0.196	0.417 **	0.336 **	−0.216	0.488 **	0.233 *	−0.176	−0.283 *
	Ni		1	0.130	0.177	−0.003	−0.112	−0.121	−0.140	−0.335 **	−0.003	0.146	0.574 **	0.343 **	0.342 **	0.076
	P			1	0.339 **	0.302 **	0.176	−0.187	0.632 **	0.255 *	0.128	−0.019	0.198	0.491 **	−0.122	0.006
	Fe				1	−0.014	0.140	−0.359 **	−0.039	0.023	0.017	0.052	0.486 **	0.405 **	0.340 **	−0.103
	Mn					1	−0.048	0.013	0.172	0.300 *	0.081	0.274 *	−0.149	0.179	−0.180	0.500 **
Root concentrations	Si						1	−0.243 *	0.465 **	0.782 **	0.482 **	−0.753 ^**^	0.169	−0.311 **	−0.701 **	−0.418 **
	Ni							1	−0.087	−0.173	−0.153	0.085	−0.767 **	−0.116	0.129	0.299 *
	P								1	0.496 **	0.377 **	−0.311 **	−0.086	−0.350 **	−0.502 **	−0.209
	Fe									1	0.352 **	−0.521 **	−0.105	−0.255 *	−0.847 **	−0.172
	Mn										1	−0.212	0.094	−0.236 *	−0.296 *	−0.707 **
TFs	Si											1	−0.046	0.330 **	0.604 **	0.364 **
	Ni												1	0.336 **	0.186	−0.231
	P													1	0.427 **	0.240 *
	Fe														1	0.200
	Mn															1

Note: The translocation factor of Si, Ni, P, Fe, and Mn in rice was calculated as shoot Ni concentration/root Ni concentration. * Correlation is significant at the 0.05 level (two-tailed). ** Correlation is significant at the 0.01 level (two-tailed).

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
