# Peer review of "Genotypic Variation in Nickel Accumulation and Translocation and Its Relationships with Silicon, Phosphorus, Iron, and Manganese among 72 Major Rice Cultivars from Jiangsu Province, China"

_ijerph, 2019, doi:10.3390/ijerph16183281_

Round 1

Reviewer 1 Report

The manuscript "Genotypic variations in nickel accumulation and translocation and their relationships with silicon, phosphorus, iron, and manganese among 72 major rice cultivars from Jiangsu Province, China", by Ya Wang and colleagues, presents an experimental study on the response of 72 rice cultivars to nickel concentration in a hydroponic environment.

The authors found significant differences in nickel accumulation and translocation among the tested cultivars, and argue that nickel uptake and transport patterns seem correlated to those found for silicon and iron rather than phosphorus and manganese. As some of the differences found were related to cultivars resulting from breeding versus their related parent cultivars, the authors finally suggest that breeding strategies should be explored to produce rice cultivars with a low tendency to accumulate nickel in the shoot.

Overall, this is a good experimental work that provides information on of a relevant number of rice cultivars in response to nickel contamination. So this work add knowledge to the topics related to the prevention of food contamination in metal polluted soils.

The manuscript is well written; the objectives and methodologies are clearly stated, the results are well illustrated, the discussion is linked to previous research and consistent with the results. Finally, I think this work is suited for publication on IJERPH, with just minor suggestions listed below:

47: "...were the most common (2.91%)..." It seems not to be a big number. Is it a percentage on the total of polluted soils or on the total soils of the area?

67-68: mercury and chromium are introduced for the first time in the manuscript, so their symbols should be given, as done for the other elements. Change to "... lead (Pb), mercury (Hg), As, Cd and chromium (Cr)".

227-228: (second and third vs third and fourth quadrants). The identification of the quadrants is not so clear. Could also refer to them as uppers/lowers, left/right.

Author Response

Response to Reviewer 1 Comments

Point 1: 47: "...were the most common (2.91%)..." It seems not to be a big number. Is it a percentage on the total of polluted soils or on the total soils of the area?

Response 1: Thank you for the valuable comment. It is a percentage on the total soils of the area. We have carefully revised the associated description in the revised manuscript (page 2, lines 47–48).

Point 2:  67-68: mercury and chromium are introduced for the first time in the manuscript, so their symbols should be given, as done for the other elements. Change to "... lead (Pb), mercury (Hg), As, Cd and chromium (Cr)".

Response 2: Followed the reviewer’s suggestion, the sentence has been changed to " … lead (Pb), mercury (Hg), As, Cd, and chromium (Cr)". (page 2, lines 67–68)

Point 3: 227-228: (second and third vs third and fourth quadrants). The identification of the quadrants is not so clear. Could also refer to them as uppers/lowers, left/right.

Response 3: Followed the reviewer’s suggestion, the "second and third quadrants" has been changed to "left upper and lower quadrants" and the "third and fourth quadrants" has been changed to "lower right and left quadrants". (page 10, lines 226–227)

Reviewer 2 Report

The manuscript “Genotypic variations in nickel accumulation and translocation and their relationships with silicon, phosphorus, iron, and manganese among 72 major rice cultivars from Jiangsu Province, China” provides a number of results that may be interesting for the scientific community. The manuscript should undergo moderate English editing, please address this during revision. The section materials and methods need of some revision. In this respect, I have appended below a list of major and minor concerns, the authors could address before publication. Therefore, without this clarification, it is difficult for me to recommend the manuscript for publication in its present form in ijerph.

Generally comment: Please, consider related papers, such as https://doi.org/10.3390/ijerph15030543, and searching in MDPI for "nickel, rice" you can find some interesting updated works which you could use to further improve and update Introduction and Discussion.

Keywords generally are not reported the same word of the title

line 40 and in all manuscript change the heavy metals with potential toxic metals or elements

Line 67 and lin 68 insert references

line 68 Cr instead Chromium

Line 76 and 78 do you have some information on Ca or Mg? and why are not considered in the manuscript

 Line 102 insert reference

Line 107 why these concentration?

Line 109 why only 3 days? (chronic effect are considered?)

Line 132 insert tukey analysis

Line 152 and 154 insert TFs and BCFs information in MandM section

Table 3 are you sure that for each sample you have just a 1 letter?

Line 249 why at this concentration?

Line 276 which As form?

Line 301 do you have some pictures?

Author Response

Response to Reviewer 2 Comments

Point 1: The manuscript “Genotypic variations in nickel accumulation and translocation and their relationships with silicon, phosphorus, iron, and manganese among 72 major rice cultivars from Jiangsu Province, China” provides a number of results that may be interesting for the scientific community. The manuscript should undergo moderate English editing, please address this during revision. The section materials and methods need of some revision. In this respect, I have appended below a list of major and minor concerns, the authors could address before publication. Therefore, without this clarification, it is difficult for me to recommend the manuscript for publication in its present form in ijerph.

Response 1: We thank the reviewer for this valuable comment. The English in this document has been checked by at least two professional editors, both native speakers of English. For a certificate, please see: http://www.textcheck.com/certificate/WLKx9e. Additionally, we have revised the section materials and methods as suggested. Please see details in the revised manuscript.

Point 2: Generally comment: Please, consider related papers, such as https://doi.org/10.3390/ijerph15030543, and searching in MDPI for "nickel, rice" you can find some interesting updated works which you could use to further improve and update Introduction and Discussion.

Response 2: We appreciate the kind suggestion. Following the reviewer’s comment, four references (i.e., 2, 18, 22, 57) have been added to the revised manuscript (page 13, line 362–page 15, line 499). Furthermore, we have carefully revised the associated discussion in the revised manuscript.

Point 3: Keywords generally are not reported the same word of the title

Response 3: Thank you for your useful comment. According to your suggestion, we have revised the Keywords. Please see details in the revised manuscript (page 1, line 31).

Point 4: line 40 and in all manuscript change the heavy metals with potential toxic metals or elements

Response 4: Followed the reviewer’s suggestion, the "heavy metals" has been changed to "potentially toxic element (PTE)" in the revised manuscript.

Point 5: Line 67 and lin 68 insert references

Response 5: Followed the reviewer’s suggestion, five references (i.e., 9-12, 14) have been inserted into the revised manuscript (page 2, line 68).

Point 6: line 68 Cr instead Chromium

Response 6: Chromium is introduced for the first time in the manuscript, so the acronym should be given. Followed the reviewer’s suggestion, the "chromium" has been changed to "chromium (Cr)". (page 2, line 68)

Point 7: Line 76 and 78 do you have some information on Ca or Mg? and why are not considered in the manuscript

Response 7: Thank you for the valuable comment. Ni exposure has been shown to decrease Ca and Mg uptake and distribution in rice (Rubio et al., 1994). In addition, Ca can reduce the toxic effects of Ni by decreasing the translocation of Ni towards the rice shoots, thereby improving rice growth (Aziz et al., 2015). Therefore, these elements are also important for alleviating the toxic effects of Ni. However, Ca analysis is difficult using ICP-MS, due to the interference with Ar. Thus, we did not analyze Ca in this study. We plan to investigate the effects of Ca and Mg (using atomic absorption spectrophotometry at another institution) on Ni accumulation and translocation by rice in different Ni-contaminated soils, and hope that there will be more opportunities to discuss this with you in the future.

Aziz, H.; Sabir, M.; Ahmad, H. R.; Aziz, T.; Zia-ur-Rehman, M.; Hakeem, K. R.; Ozturk, M. Alleviating Effect of Calcium on Nickel Toxicity in Rice. CLEAN – Soil, Air, Water 2015, 43, 901-909.

Rubio, M.I.; Escrig, I.; Martínez-Cortina, C.; López-Benet, F.J.; Sanz, A. Cadmium and nickel accumulation in rice plants. Effects on mineral nutrition and possible interactions of abscisic and gibberellic acids. Plant Growth Regul. 1994, 14, 151-157.

Point 8: Line 102 insert reference

Response 8: Followed the reviewer’s suggestion, two references (i.e., 41, 42) have been inserted into the revised manuscript (page 3, line 103).

Point 9: Line 107 why these concentration?

Response 9: Thank you for your comment. The 10 μmol L−1 Ni treatment was selected because 1) a previous study demonstrated that 10–200 μmol L−1 Ni stress can restrict the growth and photosynthetic parameters of rice seedlings (Rizwan et al., 2017); 2) in our preliminary experiment, the uptake of nutrient elements such as Si, Fe, and Mn was significantly inhibited in rice under 10 μmol L−1 Ni stress for 3 days, indicating that Ni is toxic to rice seedlings at this dose―similar results were obtained in the present study (Figure S4 and S5); and 3) although the Ni concentration can reach 26,000 mg/kg in polluted soils, its concentration in polluted surface waters or pore waters usually does not exceed 0.2 mg/L (i.e., 3.4 μmol L−1 Ni) (Yusuf et al., 2011). In conclusion, we chose 10 μmol L−1 for the Ni treatment, and rice seedlings with no Ni served (0 μmol L−1) as a control. We have added a reference (i.e., 7) pertaining to dose selection to the revised manuscript.

Rizwan, M.; Imtiaz, M.; Dai, Z.; Mehmood, S.; Adeel, M.; Liu, J.; Tu, S. Nickel stressed responses of rice in Ni subcellular distribution, antioxidant production, and osmolyte accumulation. Environ. Sci. Pollut. R. 2017, 24, 20587-20598.

Yusuf, M.; Fariduddin, Q.; Hayat, S.; Ahmad, A. Nickel: An Overview of Uptake, Essentiality and Toxicity in Plants. Bull. Environ. Contam. Toxicol. 2011, 86, 1-17.

Point 10: Line 109 why only 3 days? (chronic effect are considered?)

Response 10: Thank you very much for your comment. The nutrient solutions were renewed every 3 days, which represents a short-term experiment. Similar exposure periods were used in other studies (Astolfi et al., 2005; Chen et al., 2007; Shi et al., 2015). The chronic effects were not considered in the present study, but we will investigate the variation in Ni accumulation and translocation between the 20 low-Ni and 20 high-Ni cultivar groups identified in this study in plant pot experiments (i.e., long-term experiments), and how genotypes influence Ni uptake and transport in rice.

Astolfi, S.; Zuchi, S.; Passera, C. Effect of cadmium on H+ATPase activity of plasma membrane vesicles isolated from roots of different S-supplied maize (Zea mays L.) plants. Plant Sci. 2005, 169, 361-368.

Chen, P.-Y.; Huang, T.-L.; Huang, H.-J. Early events in the signalling pathway for the activation of MAPKs in rice roots exposed to nickel. Funct. Plant Biol. 2007, 34, 995-1001.

Shi, G.L.; Zhu, S.; Meng, J.R.; Qian, M.; Yang, N.; Lou, L.Q.; Cai, Q.S. Variation in arsenic accumulation and translocation among wheat cultivars: The relationship between arsenic accumulation, efflux by wheat roots and arsenate tolerance of wheat seedlings. J. Hazard. Mater. 2015, 190-196.

Point 11: Line 132 insert tukey analysis

Response 11: We appreciate the kind suggestion and have inserted the tukey analysis as suggested. (page 4, line 139)

Point 12: Line 152 and 154 insert TFs and BCFs information in M and M section

Response 12: Thank you for this useful suggestion. Followed your suggestion, we defined the TFs and BCFs in the materials and methods section (page 4, lines 115–118).

Point 13: Table 3 are you sure that for each sample you have just a 1 letter?

Response 13: Following the reviewer’s valuable comment, we re-analyzed the data in Table 3 using SPSS and obtained the same results. We also provide our original data in the table below.

Cultivars abbreviation

Shoot Ni concentration

(mg kg−1)

Cultivars abbreviation

Shoot Ni concentration

 (mg kg−1)

HD8

22.175

YD6

24.460

22.029

24.495

22.406

24.734

HD9

22.304

YD8

22.182

11.842

22.212

12.818

22.418

HD11

21.482

YJ2

25.984

23.014

26.597

24.953

26.536

NJ11

21.458

ZD16

18.773

23.516

20.100

23.167

20.388

WJ15

20.232

ZD88

21.044

19.087

22.053

21.168

19.766

WYJ3

24.463

ZD99

25.840

23.495

27.002

24.071

25.218

Point 14: Line 249 why at this concentration?

Response 14: We are sorry for the error in the previous statement. We confused Ni with As (0.75 mg L−1 As equivalent to 10 μmol L−1 As) because our work mainly focuses on As. In fact, the exposure concentration was 10 μmol L−1 (for Ni). A detailed check of the revised manuscript was performed to avoid similar confusing descriptions.

Point 15: Line 276 which As form?

Response 15: Both arsenite and trivalent methylarsonous acid induced PC syntheses in rice. Accordingly, the two As species have been added to the revised manuscript (page 11, lines 274–275).

Point 16: Line 301 do you have some pictures?

Response 16: We did not take pictures during the experiments. Nevertheless, we did not see an obvious reddish-brown coating on the surface of the rice roots. We believe that the supply of excess P in the culture medium (i.e., 5 mg/L P for 3 days) is responsible for the results because many studies have demonstrated that iron plaque is induced by P starvation. (Kirk and Van Du 1997; Liu et al., 2004).

Kirk, G. J. D.; Van Du, L. E. Changes in rice root architecture, porosity, and oxygen and proton release under phosphorus deficiency. New Phytol. 1997, 135, 191-200.

Liu, W. J.; Zhu, Y. G.; Smith, F. A.; Smith, S. E. Do phosphorus nutrition and iron plaque alter arsenate (As) uptake by rice seedlings in hydroponic culture? New Phytol. 2004, 162, 481-488.

Round 2

Reviewer 2 Report

All question are solved. 

I think that the manuscript could be published in present form.